# MODEL-ENSEMBLE TRUST-REGION POLICY OPTIMIZATION

**Thanard Kurutach**  **Ignasi Clavera**  **Yan Duan**  **Aviv Tamar**  **Pieter Abbeel**
Berkeley AI Research
University of California, Berkeley
Berkeley, CA 94709
`{thanard.kurutach, iclavera, rockyduan, avivt, pabbeel}@berkeley.edu`

## ABSTRACT

Model-free reinforcement learning (RL) methods are succeeding in a growing number of tasks, aided by recent advances in deep learning. However, they tend to suffer from high sample complexity which hinders their use in real-world domains. Alternatively, model-based reinforcement learning promises to reduce sample complexity, but tends to require careful tuning and, to date, it has succeeded mainly in restrictive domains where simple models are sufficient for learning. In this paper, we analyze the behavior of vanilla model-based reinforcement learning methods when deep neural networks are used to learn both the model and the policy, and we show that the learned policy tends to exploit regions where insufficient data is available for the model to be learned, causing instability in training. To overcome this issue, we propose to use an ensemble of models to maintain the model uncertainty and regularize the learning process. We further show that the use of likelihood ratio derivatives yields much more stable learning than backpropagation through time. Altogether, our approach Model-Ensemble Trust-Region Policy Optimization (ME-TRPO) significantly reduces the sample complexity compared to model-free deep RL methods on challenging continuous control benchmark tasks[1].

## 1 INTRODUCTION

Deep reinforcement learning has achieved many impressive results in recent years, including learning to play Atari games from raw-pixel inputs (Mnih et al., 2015), mastering the game of Go (Silver et al., 2016; 2017), as well as learning advanced locomotion and manipulation skills from raw sensory inputs (Levine et al., 2016a; Schulman et al., 2015; 2016; Lillicrap et al., 2015). Many of these results were achieved using *model-free* reinforcement learning algorithms, which do not attempt to build a model of the environment. These algorithms are generally applicable, require relatively little tuning, and can easily incorporate powerful function approximators such as deep neural networks. However, they tend to suffer from high sample complexity, especially when such powerful function approximators are used, and hence their applications have been mostly limited to simulated environments. In comparison, model-based reinforcement learning algorithms utilize a learned model of the environment to assist learning. These methods can potentially be much more sample efficient than model-free algorithms, and hence can be applied to real-world tasks where low sample complexity is crucial (Deisenroth and Rasmussen, 2011; Levine et al., 2016a; Venkatraman et al., 2017). However, so far such methods have required very restrictive forms of the learned models, as well as careful tuning for them to be applicable. Although it is a straightforward idea to extend model-based algorithms to deep neural network models, so far there has been comparatively fewer successful applications.

The standard approach for model-based reinforcement learning alternates between model learning and policy optimization. In the model learning stage, samples are collected from interaction with the environment, and supervised learning is used to fit a dynamics model to the observations. In the policy optimization stage, the learned model is used to search for an improved policy. The underlying assumption in this approach, henceforth termed *vanilla* model-based RL, is that with enough data,

---

[1]Videos available at: `https://sites.google.com/view/me-trpo`

the learned model will be accurate enough, such that a policy optimized on it will also perform well in the real environment.

Although vanilla model-based RL can work well on low-dimensional tasks with relatively simple dynamics, we find that on more challenging continuous control tasks, performance was highly unstable. The reason is that the policy optimization tends to exploit regions where insufficient data is available to train the model, leading to catastrophic failures. Previous work has pointed out this issue as *model bias*, i.e. (Deisenroth and Rasmussen, 2011; Schneider, 1997; Atkeson and Santamaria, 1997). While this issue can be regarded as a form of overfitting, we emphasize that standard countermeasures from the supervised learning literature, such as regularization or cross validation, are not sufficient here – supervised learning can guarantee generalization to states from the same distribution as the data, but the policy optimization stage steers the optimization exactly towards areas where data is scarce and the model is inaccurate. This problem is severely aggravated when expressive models such as deep neural networks are employed.

To resolve this issue, we propose to use an ensemble of deep neural networks to maintain model uncertainty given the data collected from the environment. During model learning, we differentiate the neural networks by varying their weight initialization and training input sequences. Then, during policy learning, we regularize the policy updates by combining the gradients from the imagined stochastic roll-outs. Each imagined step is uniformly sampled from the ensemble predictions. Using this technique, the policy learns to become robust against various possible scenarios it may encounter in the real environment. To avoid overfitting to this regularized objective, we use the model ensemble for early stopping policy training.

Standard model-based techniques require differentiating through the model over many time steps, a procedure known as backpropagation through time (BPTT). It is well-known in the literature that BPTT can lead to exploding and vanishing gradients (Bengio et al., 1994). Even when gradient clipping is applied, BPTT can still get stuck in bad local optima. We propose to use likelihood ratio methods instead of BPTT to estimate the gradient, which only make use of the model as a simulator rather than for direct gradient computation. In particular, we use Trust Region Policy Optimization (TRPO) (Schulman et al., 2015), which imposes a trust region constraint on the policy to further stabilize learning.

In this work, we propose Model-Ensemble Trust-Region Policy Optimization (ME-TRPO), a model-based algorithm that achieves the same level of performance as state-of-the-art model-free algorithms with $100\times$ reduction in sample complexity. We show that the model ensemble technique is an effective approach to overcome the challenge of model bias in model-based reinforcement learning. We demonstrate that replacing BPTT by TRPO yields significantly more stable learning and much better final performance. Finally, we provide an empirical analysis of vanilla model-based RL using neural networks as function approximators, and identify its flaws when applied to challenging continuous control tasks.

## 2 RELATED WORK

There has been a large body of work on model-based reinforcement learning. They differ by the choice of model parameterization, which is associated with different ways of utilizing the model for policy learning. Interestingly, the most impressive robotic learning applications so far were achieved using the simplest possible model parameterization, namely linear models (Bagnell and Schneider, 2001; Abbeel et al., 2006; Levine and Abbeel, 2014; Watter et al., 2015; Levine et al., 2016a; Kumar et al., 2016), where the model either operates directly over the raw state, or over a feature representation of the state. Such models are very data efficient, and allows for very efficient policy optimization through techniques from optimal control. However, they only have limited expressiveness, and do not scale well to complicated nonlinear dynamics or high-dimensional state spaces, unless a separate feature learning phase is used (Watter et al., 2015).

An alternative is to use nonparametric models such as Gaussian Processes (GPs) (Rasmussen et al., 2003; Ko et al., 2007; Deisenroth and Rasmussen, 2011). Such models can effectively maintain uncertainty over the predictions, and have infinite representation power as long as enough data is available. However, they suffer from the curse of dimensionality, and so far their applications have

been limited to relatively low-dimensional settings. The computational expense of incorporating the uncertainty estimates from GPs into the policy update also imposes an additional challenge.

Deep neural networks have shown great success in scaling up model-free reinforcement learning algorithms to challenging scenarios (Mnih et al., 2015; Silver et al., 2016; Schulman et al., 2015; 2016). However, there has been only limited success in applying them to model-based RL. Although many previous studies have shown promising results on relatively simple domains (Nguyen and Widrow, 1990; Schmidhuber and Huber, 1991; Jordan and Rumelhart, 1992; Gal et al., 2016), so far their applications on more challenging domains have either required a combination with model-free techniques (Oh et al., 2015; Heess et al., 2015; Nagabandi et al., 2017), or domain-specific policy learning or planning algorithms (Lenz et al., 2015; Agrawal et al., 2016; Pinto and Gupta, 2016; Levine et al., 2016b; Finn and Levine, 2017; Nair et al., 2017). In this work, we show that our purely model-based approach improves the sample complexity compared to methods that combine model-based and model-free elements.

Two recent studies have shown promising signs towards a more generally applicable model-based RL algorithm. Depeweg et al. (2017) utilize Bayesian neural networks (BNNs) to learn a distribution over dynamics models, and train a policy using gradient-based optimization over a collection of models sampled from this distribution. Mishra et al. (2017) learn a latent variable dynamic model over temporally extended segments of the trajectory, and train a policy using gradient-based optimization over the latent space. Both of these approaches have been shown to work on a fixed dataset of samples which are collected before the algorithm starts operating. Hence, their evaluations have been limited to domains where random exploration is sufficient to collect data for model learning. In comparison, our approach utilizes an iterative process of alternatively performing model learning and policy learning, and hence can be applied to more challenging domains. Additionally, our proposed improvements are orthogonal to both approaches, and can be potentially combined to yield even better results.

## 3 PRELIMINARIES

This paper assumes a discrete-time finite-horizon Markov decision process (MDP), defined by $(\mathcal{S}, \mathcal{A}, f, r, \rho_0, T)$, in which $\mathcal{S} \subseteq \mathbb{R}^n$ is the state space, $\mathcal{A} \subseteq \mathbb{R}^m$ the action space, $f : \mathcal{S} \times \mathcal{A} \to \mathcal{S}$ a deterministic transition function, $r : \mathcal{S} \times \mathcal{A} \to \mathbb{R}$ a bounded reward function, $\rho_0 : \mathcal{S} \to \mathbb{R}_+$ an initial state distribution, and $T$ the horizon. We denote a stochastic policy $\pi_\theta(a|s)$ as the probability of taking action $a$ at state $s$. Let $\eta(\theta)$ denote its expected return: $\eta(\theta) = \mathbb{E}_\tau[\sum_{t=0}^T r(s_t, a_t)]$, where $\tau = (s_0, a_0, \ldots, a_{T-1}, s_T)$ denotes the whole trajectory, $s_0 \sim \rho_0(.)$, $a_t \sim \pi_\theta(.|s_t)$, and $s_{t+1} = f(s_t, a_t)$ for all $t$. We assume that the reward function is known but the transition function is unknown. Our goal is to find an optimal policy that maximizes the expected return $\eta(\theta)$.

## 4 VANILLA MODEL-BASED DEEP REINFORCEMENT LEARNING

In the most successful methods of model-free reinforcement learning, we iteratively collect data, estimate the gradient of the policy, improve the policy, and then discard the data. Conversely, model-based reinforcement learning makes more extensive use of the data; it uses all the data collected to train a model of the dynamics of the environment. The trained model can be used as a simulator in which the policy can be trained, and also provides gradient information (Sutton, 1990; Deisenroth and Rasmussen, 2011; Depeweg et al., 2017; Sutton, 1991). In the following section, we describe the vanilla model-based reinforcement learning algorithm (see Algorithm 1). We assume that the model and the policy are represented by neural networks, but the methodology is valid for other types of function approximators.

### 4.1 MODEL LEARNING

The transition dynamics is modeled with a feed-forward neural network, using the standard practice to train the neural network to predict the change in state (rather than the next state) given a state and an action as inputs. This relieves the neural network from memorizing the input state, especially when the change is small (Deisenroth and Rasmussen, 2011; Fu et al., 2016; Nagabandi et al., 2017).

We denote the function approximator for the next state, which is the sum of the input state and the output of the neural network, as $\hat{f}_\phi$.

The objective of model learning is to find a parameter $\phi$ that minimizes the $L_2$ one-step prediction loss[2]:

$$\min_\phi \frac{1}{|\mathcal{D}|} \sum_{(s_t, a_t, s_{t+1}) \in \mathcal{D}} \left\| s_{t+1} - \hat{f}_\phi(s_t, a_t) \right\|_2^2 \tag{1}$$

where $\mathcal{D}$ is the training dataset that stores the transitions the agent has experienced. We use the Adam optimizer (Kingma and Ba, 2014) to solve this supervised learning problem. Standard techniques are followed to avoid overfitting and facilitate the learning such as separating a validation dataset to early stop the training, and normalizing the inputs and outputs of the neural network[3]

### 4.2 POLICY LEARNING

Given an MDP, $\mathcal{M}$, the goal of reinforcement learning is to maximize the expected sum of rewards. During training, model-based methods maintain an approximate MDP, $\hat{\mathcal{M}}$, where the transition function is given by a parameterized model $\hat{f}_\phi$ learned from data. The policy is then updated with respect to the approximate MDP. Hence, the objective we maximize is

$$\hat{\eta}(\theta; \phi) := \mathbb{E}_{\hat{\tau}}[\sum_{t=0}^{T} r(s_t, a_t)], \tag{2}$$

where $\hat{\tau} = (s_0, a_0, ...)$, $s_0 \sim \rho_0(\cdot)$, $a_t \sim \pi_\theta(\cdot|s_t)$, and $s_{t+1} = \hat{f}_\phi(s_t, a_t)$.

We represent the stochastic policy[4] as a conditional multivariate normal distribution with a parametrized mean $\mu_\theta : \mathcal{S} \to \mathcal{A}$ and a parametrized standard deviation $\sigma_\theta : \mathcal{S} \to \mathbb{R}^m$. Using the re-parametrization trick (Heess et al., 2015), we can write down an action sampled from $\pi_\theta$ at state $s$ as $\mu_\theta(s) + \sigma_\theta(s)^T \zeta$, where $\zeta \sim \mathcal{N}(0, I_m)$. Given a trajectory $\hat{\tau}$ sampled using the policy $\pi_\theta$, we can recover the noise vectors $\{\zeta_0, ..., \zeta_T\}$. Thus, the gradient of the objective $\hat{\eta}(\theta; \phi)$ can simply be estimated by Monte-Carlo methods:

$$\nabla_\theta \hat{\eta} = \mathbb{E}_{s_0 \sim \rho_0(s_0), \zeta_i \sim \mathcal{N}(0, I_m)}[\nabla_\theta \sum_{t=0}^{T} r(s_t, a_t)] \tag{3}$$

This method of gradient computation is called backpropagation through time (BPTT), which can be easily performed using an automatic differentiation library. We apply gradient clipping (Pascanu et al., 2013) to deal with exploding gradients, and we use the Adam optimizer (Kingma and Ba, 2014) for more stable learning. We perform the updates until the policy no longer improves its estimated performance $\hat{\eta}$ over a period of time (controlled by a hyperparameter), and then we repeat the process in the outer loop by using the policy to collect more data with respect to the real model[5]. The whole procedure terminates when the desired performance according to the real model is accomplished.

## 5 MODEL-ENSEMBLE TRUST-REGION POLICY OPTIMIZATION

Using the vanilla approach described in Section 4, we find that the learned policy often exploits regions where scarce training data is available for the dynamics model. Since we are improving

---

[2]We found that multi-step prediction loss did not significantly improve the policy learning results.

[3]In BPTT, maintaining large weights can result in exploding gradients; normalization relieves this effect and eases the learning.

[4]Even though for generality we present the stochastic framework of BPTT, this practice is not necessary in our setting. We found that deterministic BPTT suffers less from saturation and more accurately estimate the gradient when using a policy with a small variance or a deterministic policy.

[5]In practice, to reduce variance in policy evaluation, the initial states are chosen from the sampled trajectories rather than re-sampled from $\rho_0$.

---

**Algorithm 1** Vanilla Model-Based Deep Reinforcement Learning

---

1: Initialize a policy $\pi_\theta$ and a model $\hat{f}_\phi$.
2: Initialize an empty dataset $D$.
3: **repeat**
4:     Collect samples from the real environment $f$ using $\pi_\theta$ and add them to $D$.
5:     Train the model $\hat{f}_\phi$ using $D$.
6:     **repeat**
7:         Collect fictitious samples from $\hat{f}_\phi$ using $\pi_\theta$.
8:         Update the policy using BPTT on the fictitious samples.
9:         Estimate the performance $\hat{\eta}(\theta; \phi)$.
10:     **until** the performance stop improving.
11: **until** the policy performs well in real environment $f$.

---

the policy with respect to the approximate MDP instead of the real one, the predictions then can be erroneous to the policy's advantage. This overfitting issue can be partly alleviated by early stopping using validation initial states in a similar manner to supervised learning. However, we found this insufficient, since the performance is still evaluated using the same learned model, which tends to make consistent mistakes. Furthermore, although gradient clipping can usually resolve exploding gradients, BPTT still suffers from vanishing gradients, which cause the policy to get stuck in bad local optima (Bengio et al., 1994; Pascanu et al., 2013). These problems are especially aggravated when optimizing over long horizons, which is very common in reinforcement learning problems.

We now present our method, Model-Ensemble Trust-Region Policy Optimization (ME-TRPO). The pseudocode is shown in Algorithm 2. ME-TRPO combines three modifications to the vanilla approach. First, we fit a set of dynamics models $\{f_{\phi_1}, \ldots, f_{\phi_K}\}$ (termed a *model ensemble*) using the same real world data. These models are trained via standard supervised learning, as described in Section 4.1, and they only differ by the initial weights and the order in which mini-batches are sampled. Second, we use Trust Region Policy Optimization (TRPO) to optimize the policy over the model ensemble. Third, we use the model ensemble to monitor the policy's performance on validation data, and stops the current iteration when the policy stops improving. The second and third modifications are described in detail below.

**Policy Optimization.** To overcome the issues with BPTT, we use likelihood-ratio methods from the model-free RL literature. We evaluated using Vanilla Policy Gradient (VPG) (Peters and Schaal, 2006), Proximal Policy Optimization (PPO) (Schulman et al., 2017), and Trust Region Policy Optimization (TRPO) (Schulman et al., 2015). The best results were achieved by TRPO. In order to estimate the gradient, we use the learned models to simulate trajectories as follows: in every step, we randomly choose a model to predict the next state given the current state and action. This avoids the policy from overfitting to any single model during an episode, leading to more stable learning.

**Policy Validation.** We monitor the policy's performance using the $K$ learned models. Specifically, we compute the ratio of models in which the policy improves:

$$\frac{1}{K} \sum_{k=1}^{K} \mathbb{1}[\hat{\eta}(\theta_{new}; \phi_k) > \hat{\eta}(\theta_{old}; \phi_k)]. \tag{4}$$

The current iteration continues as long as this ratio exceeds a certain threshold. In practice, we validate the policy after every 5 gradient updates and we use 70% as the threshold. If the ratio falls below the threshold, a small number of updates is tolerated in case the performance improves, the current iteration is terminated. Then, we repeat the overall process of using the policy to collect more real-world data, optimize the model ensemble, and using the model ensemble to improve the policy. This process continues until the desired performance in the real environment is reached.

The model ensemble serves as effective regularization for policy learning: by using the model ensemble for policy optimization and validation, the policy is forced to perform well over a vast number of possible alternative futures. Even though any of the individual models can still incur model bias, our experiments below suggest that combining these models yields stable and effective policy improvement.

---

**Algorithm 2** Model Ensemble Trust Region Policy Optimization (ME-TRPO)

---

1: Initialize a policy $\pi_\theta$ and all models $\hat{f}_{\phi_1}, \hat{f}_{\phi_2}, ..., \hat{f}_{\phi_K}$.
2: Initialize an empty dataset $\mathcal{D}$.
3: **repeat**
4:     Collect samples from the real system $f$ using $\pi_\theta$ and add them to $D$.
5:     Train all models using $\mathcal{D}$.
6:     **repeat**                                             $\triangleright$ Optimize $\pi_\theta$ using all models.
7:         Collect fictitious samples from $\{\hat{f}_{\phi_i}\}_{i=1}^K$ using $\pi_\theta$.
8:         Update the policy using TRPO on the fictitious samples.
9:         Estimate the performances $\hat{\eta}(\theta; \phi_i)$ for $i = 1, ..., K$.
10:     **until** the performances stop improving.
11: **until** the policy performs well in real environment $f$.

---

## 6 EXPERIMENTS

We design the experiments to answer the following questions:

1. How does our approach compare against state-of-the-art methods in terms of sample complexity and final performance?
2. What are the failure scenarios of the vanilla algorithm?
3. How does our method overcome these failures?

We also provide in the Appendix D an ablation study to characterize the effect of each component of our algorithm.

### 6.1 ENVIRONMENTS

To answer these questions, we evaluate our method and various baselines over six standard continuous control benchmark tasks (Duan et al., 2016; Hesse et al., 2017) in Mujoco (Todorov et al., 2012): Swimmer, Snake, Hopper, Ant, Half Cheetah, and Humanoid, shown in Figure 1. The details of the tasks can be found in Appendix A.2.

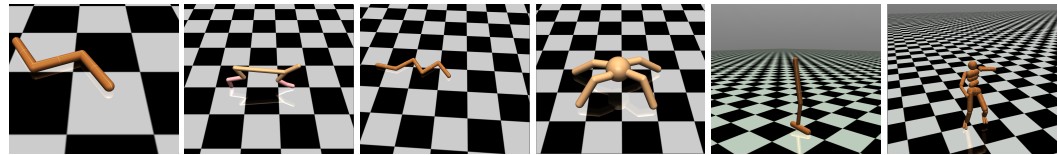

Figure 1: Mujoco environments used in our experiments. Form left to right: Swimmer, Half Cheetah, Snake, Ant, Hopper, and Humanoid.

### 6.2 COMPARISON TO STATE-OF-THE-ART

We compare our method with the following state-of-the-art reinforcement learning algorithms in terms of sample complexity and performance: Trust Region Policy Optimization (TRPO) (Schulman et al., 2015), Proximal Policy Optimization (PPO) (Schulman et al., 2017), Deep Deterministic Policy Gradient (DDPG) (Lillicrap et al., 2015), and Stochastic Value Gradient (SVG) (Heess et al., 2015).

The results are shown in Figure 2. Prior model-based methods appear to achieve worse performance compared with model-free methods. In addition, we find that model-based methods tend to be difficult to train over long horizons. In particular, SVG(1), not presented in the plots, is very unstable in our experiments. While SVG($\infty$) is more stable, it fails to achieve the same level of performance as model-free methods. In contrast, our proposed method reaches the same level of performance as model-free approaches with $\approx 100\times$ less data. To the best of our knowledge, it is the first purely model-based approach that can optimize policies over high-dimensional motor-control tasks such as Humanoid. For experiment details please refer to Appendix A.

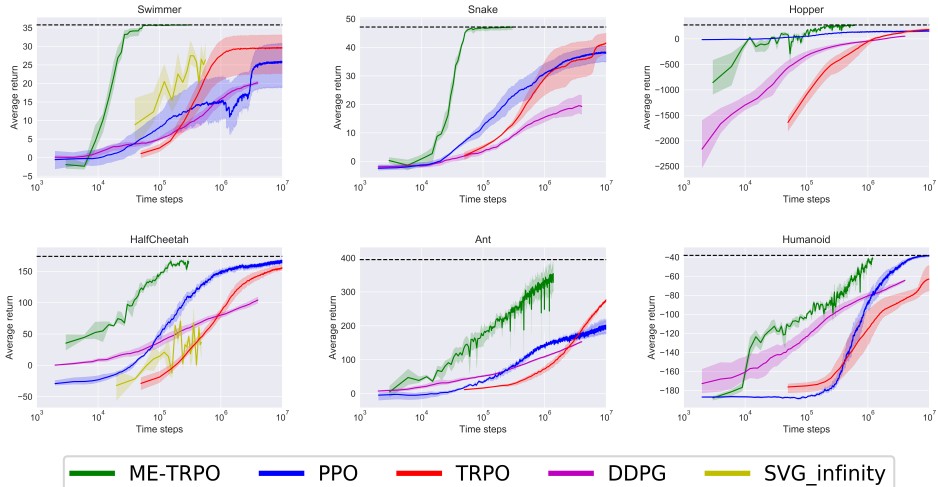

Figure 2: Learning curves of our method versus state-of-the-art methods. The horizontal axis, in log-scale, indicates the number of time steps of real world data. The vertical axis denotes the average return. These figures clearly demonstrate that our proposed method significantly outperforms other methods in comparison (best viewed in color).

## 6.3 FROM VANILLA TO ME-TRPO

In this section we explain and quantify the failure cases of vanilla model-based reinforcement learning, and how our approach overcomes such failures. We analyze the effect of each of our proposed modifications by studying the learning behavior of replacing BPTT with TRPO in vanilla model-based RL using just a single model, and then the effect of using an ensemble of models.

As discussed above, BPTT suffers from exploding and vanishing gradients, especially when optimizing over long horizons. Furthermore, one of the principal drawbacks of BPTT is the assumption that the model derivatives should match that of the real dynamics, even though the model has not been explicitly trained to provide accurate gradient information. In Figure 3 we demonstrate the effect of using policy gradient methods that make use of a score function estimator, such as VPG and TRPO, while using a single learned model. The results suggest that in comparison with BPTT, policy gradient methods are more stable and lead to much better final performance. By using such model-free algorithms, we require less information from the learned model, which only acts as a simulator. Gradient information through the dynamics model is not needed anymore to optimize the policy.

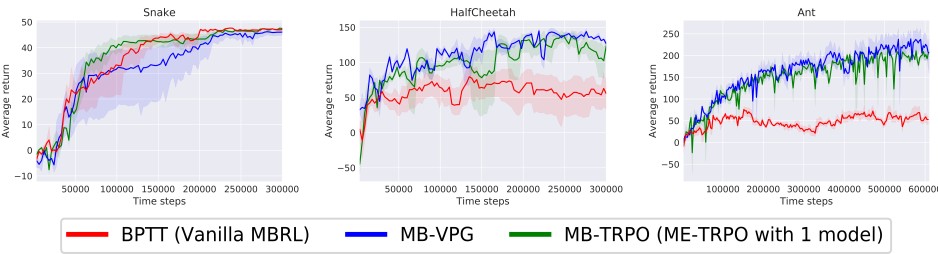

Figure 3: Comparison among different policy optimization techniques with one model. Using TRPO for model-based optimization leads to the best policy learning across the different domains (Best viewed in color).

However, while replacing BPTT by TRPO helps optimization, the learned policy can still suffer from model bias. The learning procedure tends to steer the policy towards regions where it has rarely visited, so that the model makes erroneous predictions to its advantage. The estimated performances

of the policy often end up with high rewards according to the learned model, and low rewards according to the real one (see Appendix B for further discussion). In Figure 4, we analyze the effect of using various numbers of ensemble models for sampling trajectories and validating the policy's performance. The results indicate that as more models are used in the model ensemble, the learning is better regularized and the performance continually improves. The improvement is even more noticeable in more challenging environments like HalfCheetah and Ant, which require more complex dynamics models to be learned, leaving more room for the policy to exploit when model ensemble is not used.

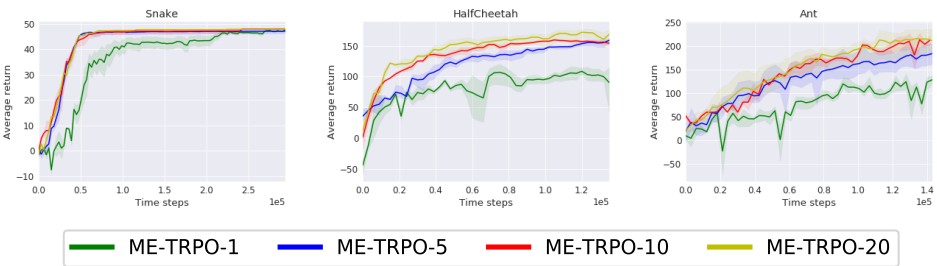

Figure 4: Comparison among different number of models that the policy is trained on. TRPO is used for the policy optimization. We illustrate the improvement when using 5, 10 and 20 models over a single model (Best viewed in color).

## 7 DISCUSSION

In this work, we present a simple and robust model-based reinforcement learning algorithm that is able to learn neural network policies across different challenging domains. We show that our approach significantly reduces the sample complexity compared to state-of-the-art methods while reaching the same level of performance. In comparison, our analyses suggests that vanilla model-based RL tends to suffer from model bias and numerical instability, and fails to learn a good policy. We further evaluate the effect of each key component of our algorithm, showing that both using TRPO and model ensemble are essential for successful applications of deep model-based RL. We also confirm the results of previous work (Deisenroth and Rasmussen, 2011; Depeweg et al., 2017; Gal et al., 2016) that using model uncertainty is a principled way to reduce model bias.

One question that merits future investigation is how to use the model ensemble to encourage the policy to explore the state space where the different models disagree, so that more data can be collected to resolve their disagreement. Another enticing direction for future work would be the application of ME-TRPO to real-world robotics systems.

ACKNOWLEDGEMENT

The authors thank Stuart Russell, Abishek Gupta, Carlos Florensa, Anusha Nagabandi, Haoran Tang, and Gregory Kahn for helpful discussions and feedbacks. T. Kurutach has been supported by ONR PECASE grant N000141612723, I. Clavera has been supported by La Caixa Fellowship, Y. Duan has been supported by Huawei Fellowship, and A. Tamar has been supported by Siemens Fellowship.

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

## A  EXPERIMENT DETAILS

### A.1  MODEL-ENSEMBLE TRUST-REGION POLICY OPTIMIZATION

Our algorithm can be broken down into three parts: data collection, model learning, and policy learning. We describe the numerical details for each part below.

#### A.1.1  DATA COLLECTION

In each outer iteration, we use the stochastic policy to collect 3000 timesteps of real world data for every environment, except Humanoid in which we collect 6000 timesteps. At the beginning of every roll-out we sample the policy standard deviation randomly from $\mathcal{U}[0.0, 3.0]$, and we keep the value fixed throughout the episode. Furthermore, we perturb the policy's parameters by adding white Gaussian noise with standard deviation proportional to the absolute difference between the current parameters and the previous one (Plappert et al., 2018; Fortunato et al., 2018). Finally, we split the collected data using a 2-to-1 ratio for training and validation datasets.

#### A.1.2  MODEL LEARNING

We represent the dynamics model with a 2-hidden-layer feed-forward neural network with hidden sizes 1024-1024 and ReLU nonlinearities. We train the model with the Adam optimizer with learning rate 0.001 using a batch size of 1000. The model is trained until the validation loss has not decreased for 25 passes over the entire training dataset (we validate the training every 5 passes).

#### A.1.3  POLICY LEARNING

We represent the policy with a 2-hidden-layer feed-forward neural network with hidden sizes 32-32 and tanh nonlinearities for all the environments, except Humanoid, in which we use the hidden sizes 100-50-25. The policy is trained with TRPO on the learned models using initial standard deviation 1.0, step size $\delta_{KL}$ 0.01, and batch size 50000. If the policy fails the validation for 25 updates (we do the validation every 5 updates), we stop the learning and repeat the overall process.

### A.2  ENVIRONMENT DETAILS

The environments we use are adopted from rllab (Duan et al., 2016). The reward functions $r_t(s_t, a_t)$ and optimization horizons are described below:

| Environments | Reward functions | Horizon |
|---|---|---|
| Swimmer | $s_t^{vel}$ - $0.005\|a_t\|_2^2$ | 200 |
| Snake | $s_t^{vel}$ - $0.005\|a_t\|_2^2$ | 200 |
| Hopper | $s_t^{vel}$ - $0.005\,\|a_t\|_2^2$ 
 $-10\max(0.45 - s_t^{height}, 0)$ 
 - $10\sum(\max(s_t - 100, 0))$ | 100 |
| Half Cheetah | $s_t^{vel}$ - $0.05\|a_t\|_2^2$ | 100 |
| Ant | $s_t^{vel}$ - $0.005\|a_t\|_2^2 + 0.05$ | 100 |
| Humanoid | $(s_t^{head} - 1.5)^2 + \|a_t\|_2^2$ | 100 |

Note that in Hopper we relax the early stopping criterion to a soft constraint in reward function, whereas in Ant we early stop when the center of mass long z-axis is outside [0.2, 1.0] and have a survival reward when alive.

The state in each environment is composed of the joint angles, the joint velocities, and the cartesian position of the center of mass of a part of the simulated robot. We are not using the contact information, which make the environments effectively POMDPs in Half Cheetah, Ant, Hopper and Humanoid. We also eliminate the redundancies in the state space in order to avoid infeasible states in the prediction.

### A.2.1 BASELINES

In Section 6.2 we compare our method against TRPO, PPO, DDPG, and SVG. For every environment we represent the policy with a feed-forward neural network of the same size, horizon, and discount factor as the ones specified in the Appendix A.1.3. In the following we provide the hyper-parameters details:

**Trust Region Policy Optimization (Schulman et al., 2016).** We used the implementation of Duan et al. (2016) with a batch size of 50000, and we train the policies for 1000 iterations. The step size $\delta_{KL}$ that we used in all the experiments was of 0.05.

**Proximal Policy Optimization (Schulman et al., 2017).** We referred to the implementation of Hesse et al. (2017). The policies were trained for $10^7$ steps using the default hyper-parameters across all tasks.

**Deep Deterministic Policy Gradient (Lillicrap et al., 2015).** We also use the implementation of Hesse et al. (2017) using a number epochs of 2000, the rest of the hyper-parameters used were the default ones.

**Stochastic Value Gradient (Heess et al., 2015).** We parametrized the dynamics model as a feed-forward neural network of two hidden layers of 512 units each and ReLU non-linearities. The model was trained after every episode with the data available in the replay buffer, using the Adam optimizer with a learning rate of $10^{-4}$, and batch size of 128. We additionally clipped the gradient we the norm was larger than 10.

## B OVERFITTING

We show that replacing the ensemble with just one model leads to the policy overoptimization. In each outer iteration, we see that at the end of the policy optimization step the estimated performance increases while the real performance is in fact decreasing (see figure 5).

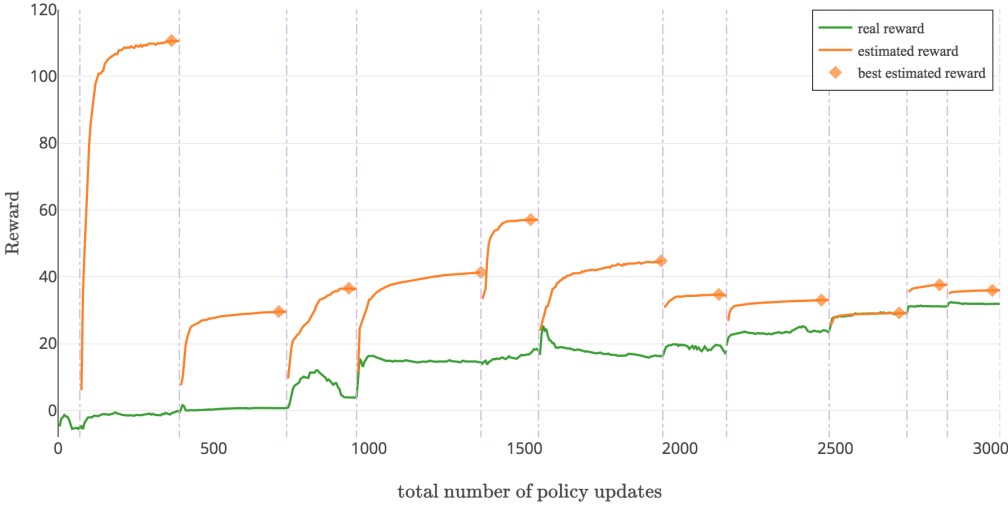

Figure 5: Predicted and real performances during the training process using our approach with one model instead of an ensemble in Swimmer. The policy overfits to the dynamics model which degrades the real performance.

## C REAL-TIME COMPLEXITY

We provide wall clock time for the ME-TRPO results from figure 2 in the table below:

| Environments | Run time (in 1000s) |
|---|---|
| Swimmer | $35.3 \pm 3.1$ |
| Snake | $60.8 \pm 4.7$ |
| Hopper | $183.6 \pm 10.2$ |
| Half Cheetah | $103.7 \pm 5.2$ |
| Ant | $395.2 \pm 67.1$ |
| Humanoid | $362.1 \pm 20.5$ |

These experiments were performed on Amazon EC2 using 1 NVIDIA K80 GPU, 4 vCPUs, and 61 GB of memory.

Note that the majority of run time is spent on training model ensemble. However, our algorithm allows this to be simply parallelized across multiple GPUs. This could potentially yield multiple-fold speed-up from our results.

## D  ABLATION STUDY

We further provide a series of ablation experiments to characterize the importance of the two main regularization components of our algorithm: the ensemble validation and the ensemble sampling techniques. In these experiments, we make only one change at a time to ME-TRPO with 5 models.

### D.1  ENSEMBLE SAMPLING METHODS

We explore several ways to simulate the trajectories from the model ensemble. At a current state and action, we study the effect of simulating the next step given by: (1) sampling randomly from the different models (step_rand), (2) a normal distribution fitted from the predictions (model_mean_std), (3) the mean of the predictions (model_mean), (4) the median of the predictions (model_med), (5) the prediction of a fixed model over the entire episode (i.e., equivalent to averaging the gradient across all simulations) (eps_rand), and (6) sampling from one model (one_model).

The results in Figure 6 provide evidence that using the next step as the prediction of a randomly sampled model from our ensemble is the most robust method across environments. In fact, using the median or the mean of the predictions does not prevent overfitting; this effect is shown in the HalfCheetah environment where we see a decrease of the performance in latter iteration of the optimization process. Using the gradient average (5) also provides room for the policy to overfit to one or more models. This supports that having an estimate of the model uncertainty, such as in (1) and (2), is the principled way to avoid overfitting the learned models.

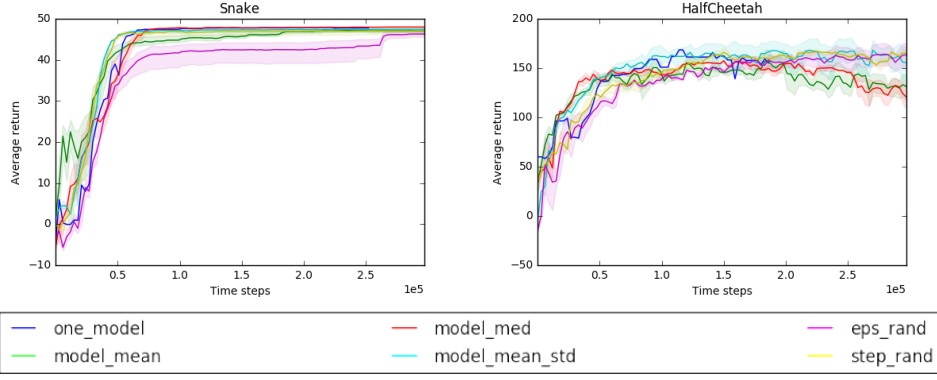

Figure 6: Comparison among different sampling techniques for simulating roll-outs. By sampling each step from a different model, we prevent overfitting and enhance the learning performance (Best viewed in color).

### D.2 ENSEMBLE VALIDATION

Finally, we provide a study of the different ways for validating the policy. We compare the following techniques: (1) using the real performance (i.e., using an oracle) (real), (2) using the average return in the trpo roll-outs (trpo_mean), (3) stopping the policy after 50 gradient updates (no_early_50), (4) or after 5 gradient updates (no_early_5), (5) using one model to predict the performances (one_model), and (6) using an ensemble of models (ensemble). The experiments are designed to use the same number of models and hyper-parameters for the other components of the algorithm.

In Figure 7 we can see the effectiveness of each approach. It is noteworthy that having an oracle of the real performance is not the best approach. Such validation is over-cautious, and does not give room for exploration resulting in a poor trained dynamics model. Stopping the gradient after a fixed number of updates results in good performance if the right number of updates is set. This burdens the hyper-parameter search with one more hyper-parameter. On the other hand, using the ensemble of models has good performance across environments without adding extra hyper-parameters.

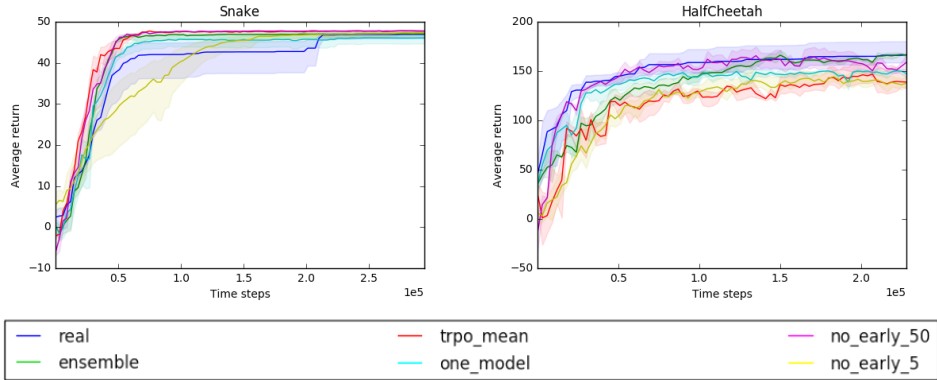

Figure 7: Comparison among validation techniques. The ensemble of models yields to good performance across environments (Best viewed in color).

