# OpenReview forum: "Model-Ensemble Trust-Region Policy Optimization"
_ICLR.cc/2018/Conference — Accept (Poster)_

### Official Review · AnonReviewer1 · 2017-11-26
**Solid contribution in relevant field**

**Rating:** 7
**Confidence:** 4

**Review:**

Summary:
The paper proposes to use ensembles of models to overcome a typical problem when training on a learned model: That the policy learns to take advantage of errors of the model.
The models use the same training data but are differentiated by a differente parameter initialization and by training on differently drawn minibatches.
To train the policy, at each step the next state is taken from an uniformly randomly drawn model.
For validation the policy is evaluated on all models and training is stopped early if it doesn't improve on enough of them.

While the idea to use an ensemble of deep neural networks to estimate their uncertainty is not new, I haven't seen it yet in this context. They successfully show in their experiments that typical levels of performance can be achieved using much less samples from the real environment.

The reduction in required samples is over an order of magnitude for simple environments (Mujoco Swimmer). However, (as expected for model based algorithms) both the performance as well as the reduction in sample complexity gets worse with increasing complexity of the environment. It can still successfully tackle the Humanoid Mujoco task but my guess is that that is close to the upper limit of this algorithm?

Overall the paper is a solid and useful contribution to the field.

*Quality:*
The paper is clearly shows the advantage of the proposed method in the experimental section where it compares to several baselines (and not only one, thank you for that!).

Things which in my opinion aren't absolutely required in the paper but I would find interesting and useful (e.g. in the appendix) are:
1. How does the runtime (e.g. number of total samples drawn from both the models and the real environment, including for validation purpuses) compare?
From the experiments I would guess that MB-TRPO is about two to three orders of magnitude slower, but having this information would be useful.
2. For more complex environments it seems that training is becoming less stable and performance degradates, especially for the Humanoid environment. A plot like in figure 4 (different number of models) for the humanoid environment could be interesting? Additionally maybe a short discussion where the major problem for further scaling lies? For example: Expressiveness of the models? Required number of models / computation feasibility? Etc... This is not necessarily required for the paper but would be interesting.

*Originality & Significance:*
As far as I can tell, none of the fundamental ideas are new. However, they are combined in an interesting, novel way that shows significant performance improvements.
The problem the authors tackle, namely learning a deep neural network model for model based RL, is important and relevant. As such, the paper contributes to the field and should be accepted.

*Smaller questions and notes:*
- Longer training times for MB-TRPO, in particular for Ant and Humanoid would have been intersting if computationionally feasibly.
- Could this in principle be used with Q-learning as well (instead of TRPO) if the action space is discrete? Or is there an obvious reason why not that I am missing?

---

### Official Review · AnonReviewer2 · 2017-11-27
**well-written, good experiments, but limited novelty, doubts about time-complexity, and use of ensembles over BNNs**

**Rating:** 6
**Confidence:** 3

**Review:**

The authors combine an ensemble of DNNs as model for the dynamics with TRPO. The ensemble is used in two steps:
First to collect imaginary roll-outs for TRPO and secondly to estimate convergence of the algorithm. The experiments indicate superior performance over the baselines.

The paper is well-written and the experiments indicate good results. However, idea of using ensembles in the context of
(model-based) RL  is not novel, and it comes at the cost of time complexity.  Therefore, the method should utilize
the advantage an ensemble provides to its full extent.
The main strength of an ensemble is to provide lower test error, but also some from of uncertainty estimate given by the spread of the predictions. The authors mainly utilize the first, but to a lesser extent the second advantage (the imaginary roll-outs will  utilize the spread to generate possible outcomes). Ideally the exploration should also be guided by the uncertainty (such as VIME).

Related, what where the arguments in favor of an ensemble compared to Bayesian neural networks (possibly even as simple as using MH-dropout)? BNNs provide a stronger theoretical justification that the predictive uncertainty is meaningful.

Can the authors comment on the time-complexity of the proposed methods compared to the baselines? In Fig. 2  the x-axis is the time step of the  real data. But I assume it took a different amount of time for each method to reach step t. The same argument can be made for Fig. 4. It seems here that in snake the larger ensembles reach convergence the quickest, but I expect this effect to be reversed when considering actual training time.

In total I think this paper can provide a useful addition to the literature.  However, the proposed approach does not have strong novelty and I am not fully convinced if the additional burden on time complexity outweighs the improved performance.

Minor:  In Sec. 2: "Both of these approaches assume a fixed dataset of samples which are collected
before the algorithm starts operating."  This is incorrect, while these methods consider the domain of fixed datasets, the algorithms themselves are not limited to this context.

---

### Official Review · AnonReviewer3 · 2017-11-28
**A nice baseline in the epic model-based vs. model-free battle**

**Rating:** 7
**Confidence:** 5

**Review:**

This paper presents a simple model-based RL approach, and shows that with a few small tweaks to more "typical" model-based procedures, the methods can substantially outperform model-free methods on continuous control tasks.  In particular, the authors show that by 1) using an ensemble of models instead of a single models, 2) using TRPO to optimize the policy based upon these models (rather that analytical gradients), and 3) using the model ensemble to validate when to stop policy optimization, then a simple model-based approach actually can outperform model-free methods.

Overall, I think this is a nice paper, and worth accepting.  There is very little actually new here, of course: the actual model-based method is entirely standard except with the additions above (which are also all fairly standard approaches in isolation).  But at a higher level, the fact that such simple model-based approaches work better than somewhat complex model free approaches actually is the point of the paper to me.  While the general theme of model-based RL outperforming model-free RL is not new (Atkeson and Santamaria (1997) comes to a similar conclusion) its good to see this same pattern demonstrated "officially" on modern RL benchmarks, especially since the _completely_ naive strategy of using a single model and more standard policy optimization doesn't perform as well.

Naturally, there is some question as to whether the work here is novel enough to warrant publication, but I think the overall message of the paper is strong enough to overcome fairly minimal contribution from an algorithmic perspective.  I did also have a few general concerns that I think could be discussed with a bit more detail in the paper:
1) The choice of this particular model ensemble to represent uncertainty seems rather ad-how.  Why is it sufficient to simply learn N models with different initial weights?  It seems that the likely cause for this is that the random initial weights may lead to very different behavior in the unobserved parts of the space (i.e., portions of the state space where we have no samples), and thus.  But it seems like there are much more principled ways of overcoming this same problem, e.g. by using an actual Bayesian neural net, directly modeling uncertainty in the forward model, or using generative model approaches.  There's some discussion of this point in the introduction, but I think a bit more explanation about why the model ensemble is expected to work well for this purpose.
2) Likewise, the fact the TRPO outperforms more standard gradient methods is somewhat surprising.  How is the model ensemble being treated during BPTT?  In the described TRPO method, the authors use a different model at each time step, sampling uniformly.  But it seems like a single model is used for each rollout in the proposed BPTT method?  If so, it's not surprising that this approach performs worse.  But it seems like one could backprop through the different per-timestep models just as easily, and it would remove one additional source of difference between the two settings.

---

### Public Comment · (anonymous) · 2017-11-15
**policy learning method choice**

I was wondering what is the difference in results between using TRPO and PPO for policy learning. PPO seems to be more stable and sample efficient than TRPO.

---

> ### Author Response · Authors · 2017-11-15
> **Re: policy learning method choice**
>
> Thank you for your comment. Since we do not care about the fictitious sample complexity, we do not find that PPO consistently improves the real sample complexity. We also noticed that the hyperparameters in PPO are more tricky to tune at least in the model-based setting, whereas TRPO can work well out-of-the-box.

---

> > ### Public Comment · (anonymous) · 2017-11-15
> > **Thanks for your reply**
> >
> > Thank you for your explanation.

---

> > ### Public Comment · (anonymous) · 2017-11-19
> > **replicating results**
> >
> > I am trying to replicate your results, it is unclear to me what exactly the value of standard derivation for perturbing policy parameters is(see A.1.1DATA COLLECTION), there only states that it is proportional to the absolute difference. Another question in A.1.1 is that 'we split the collected data using a 2-to-1 ratio for training and validation datasets' while in algorithm.2 it seems that you collect all previous data in D and use it to train, do you split current collected data or dataset D.  Could you please help clarify this? BTW, would you like to open source  codebase in the future?

---

> > > ### Author Response · Authors · 2017-11-20
> > > **Re: replicating results**
> > >
> > > Thank you for your questions. The standard deviation of parameter noise is proportional to the difference between the current parameters and the previous ones. We use 3.0 for the proportional ratio.
> > > The dataset D consists of both training and validation sets. After new data are collected, we split them and put them into each set. We plan to release the codebase in the future.

---

> > > > ### Public Comment · (anonymous) · 2017-11-20
> > > > **Re: Re: replicating results**
> > > >
> > > > Thanks, I am wondering dataset D consist of only on-policy data or all previous collected data, could you clarify this? Another question is what is the structure of f(s, a) is, you only mentioned it has hidden sizes 1024-1024 and ReLU nonlinearities.

---

> > > > > ### Author Response · Authors · 2017-11-22
> > > > > **Re: replicating results**
> > > > >
> > > > > D contains all the data collected so far. The model is a feed-forward neural network with two hidden layers.

---

> > > > > > ### Public Comment · (anonymous) · 2017-11-22
> > > > > > **Replicating results**
> > > > > >
> > > > > > Thanks, hmmm, I know it is a MLP but how do you deal with s and a since they have different dimensions.

---

> > > > > > > ### Author Response · Authors · 2017-11-22
> > > > > > > **Re: replicating results**
> > > > > > >
> > > > > > > We feed both s and a into the input. They don't need to have the same dimension.

---

> > > > ### Public Comment · (anonymous) · 2017-11-30
> > > > **question about experiments running time**
> > > >
> > > > I am wondering how long your method needs to run when using 10 dynamics in some experiments such as Swimmer-v1 and Humanoid-v1?
> > > > Could you please tell us and also provide a description of your experiment facilities.

---

### Public Comment · (anonymous) · 2017-11-16
**Relationship to EPOpt?**

This paper from last year's ICLR also considers an ensemble of models and uses TRPO to find a robust policy with reduced sample complexity. https://arxiv.org/abs/1610.01283

Can you comment in the connections? It seems very relevant, and this line of work should be cited and discussed in the paper. Domain randomization based approaches also fall under this bucket. The only difference I see is that EPOpt uses a physics based model representation whereas DNN models are used here. However, this difference is extremely minor, since the way the model is updated is identical in both -- gradient descent or MAP (in Bayesian case). Is the method proposed here simply EPOpt with DNN function approximator for the model?

---

> ### Author Response · Authors · 2017-11-17
> **Re: Relationship to EPOpt?**
>
> Thank you for your question. Here is the explanation.
>
> The use of model uncertainty to deal with the discrepancy between the model and real world dynamics is a mature idea that dates back to much earlier than the EPOpt paper. Robust MDPs [1,2], ensemble methods [3], Bayesian approaches such as PILCO [4], among others, all share the same idea - use data to estimate some form of model uncertainty, and find a policy that works well against all models in the uncertainty set, with the hope that this policy will also work well in the real world model.
>
> The differences between these works is in the models that they learn - discrete models in [1,2], Gaussian processes in [4], and physical models with a small set of parameters in [3] and EPOpt.
>
> To date, learning dynamics models with deep neural networks for non-trivial tasks has been notoriously hard. For example, in [5], Nagabandi et al. proposed to use model-free fine tuning after model-based training, and in [6] Gal et al. showed primary results on cartpole.
>
> The promise in using neural network models is their expressiveness -- which can scale up to complex dynamics for which Gaussian processes are not applicable, while writing down an analytical physics model is too challenging. This would be the case, for example, in a real-world robot that needs to handle deformable objects. For such domains, writing down a parametric physical model, as was done in EPOpt, can be problematic.
>
> So, while the main difference with prior work on model uncertainty is in our DNN model, our contribution is in showing that, for the first time, such expressive models can be used to solve challenging control tasks. This should not be waived off as a minor difference, as getting DNNs to learn useful dynamics models has been the focus of many recent studies [5,6,7].
>
> Intuitively, the challenge in model based RL with DNNs is that as the models become more expressive, the harder it becomes to control generalization errors, which the policy optimization tends to exploit (thus leading to a failure in the real-world execution). To put things in numbers, our DNN models have thousands of tunable parameters. The models in the EPOpt paper had at most 4.
>
> [1] Bagnell, J.A., Ng, A.Y. and Schneider, J.G., 2001. Solving uncertain Markov decision processes.
> [2] Nilim, A. and El Ghaoui, L., 2005. Robust control of Markov decision processes with uncertain transition matrices. Operations Research, 53(5), pp.780-798.
> [3] Mordatch, I., Lowrey, K. and Todorov, E., 2015, September. Ensemble-CIO: Full-body dynamic motion planning that transfers to physical humanoids. In Intelligent Robots and Systems (IROS), 2015 IEEE/RSJ International Conference on(pp. 5307-5314). IEEE.
> [4] Deisenroth, M. and Rasmussen, C.E., 2011. PILCO: A model-based and data-efficient approach to policy search. In Proceedings of the 28th International Conference on machine learning (ICML-11) (pp. 465-472).
> [5] Nagabandi, A., Kahn, G., Fearing, R.S. and Levine, S., 2017. Neural network dynamics for model-based deep reinforcement learning with model-free fine-tuning. arXiv preprint arXiv:1708.02596.
> [6] Gal, Y., McAllister, R.T. and Rasmussen, C.E., 2016, April. Improving PILCO with bayesian neural network dynamics models. In Data-Efficient Machine Learning workshop (Vol. 951, p. 2016).
> [7] Heess, N., Wayne, G., Silver, D., Lillicrap, T., Erez, T. and Tassa, Y., 2015. Learning continuous control policies by stochastic value gradients. In Advances in Neural Information Processing Systems (pp. 2944-2952).

---

> > ### Public Comment · (anonymous) · 2017-11-20
> > **Relationship to EPOpt**
> >
> > Thanks for the response. Your point about model expressiveness is well taken. However, the justifications still seem inadequate primarily because the chosen tasks and results do not adequately represent your premise.
> >
> > On the question of physics based models vs DNN models -- this is still wide open. While DNN models are more expressive they might require orders of magnitude more samples to train. In addition, ability to generalize to states that have not been sufficiently visited is also hard to reason -- it is precisely for this reason that DAGGER and related approaches aggregate the data sets and slowly update the policy. Physics based models suffer less from this issue due to having the right priors. Thus, while it might be possible that in the large sample case expressive models might win owing to their capacity, these are not the regimes typical robotics problems operate in. Of course, if actual hardware results were presented, your proposed motivations and premise would have been well justified, but this is not the case.
> >
> > On the difficulty on model learning -- it is not clear if there have been negative results in recent literature. The proposed way to learn the model is not very different from the vanilla approach of DAGGER for System ID -- are there any particular differences from simply aggregating the data sets and minimizing L2 loss? My guess is that not many people actually tried or implemented this correctly. An analysis of how accurate the learned model is would also be very revealing. For example, predict next state using learned model and visualize this in the simulator -- do we see smooth transitions or are the state transitions physically implausible? I could imagine that the learned models are not very accurate for prediction, and hence was not pursued rigorously. However, the models might be sufficient for policy improvement -- if this is the case, it might be an interesting insight.
> >
> > I should emphasize that my intention is not to be overly critical of the work. Model based RL is indeed a promising approach and your paper is one of the first to actually show good positive results with it in recent times. However, my concern is that many related works have been ignored, and the method is somewhat oversold. This is of course not entirely negative -- if a simple method works well but has been ignored by the community, it is worth pointing this out.

---

> > > ### Author Response · Authors · 2017-11-23
> > > **Re: Relationship to EPOpt**
> > >
> > > Thank you for your response.
> > >
> > > With the goal of keeping the paper at an 8-page length, we only provided a limited literature survey, comparing to what we thought to be the most related recent work on model based RL. We would be happy to include a comprehensive section on related work in the final version, including a comparison to DAgger and other SysId methods.
> > >
> > > We are also starting to investigate a real robot implementation of our ideas. Naturally, there are many challenges in getting model based RL to work on real hardware, and we intend this to be part of a different publication. Because of the hardware difficulty, a lot of recent work have also tested their algorithms on challenging OpenAI benchmark and open source their code. We found this to be useful to quickly and fairly compare our algorithm to other state-of-the-art methods.
> > >
> > > A difficulty in model learning in model-based setting is the coupling between the model and the policy, i.e., in order to learn a good model we need to learn a good policy and vice versa. In the paper we show that using a single model doesn't allow us to learn a good policy due to overfitting. As a result, the model cannot provide accurate enough predictions for optimizing the policy as shown - see the learning curve in figure 4. We attach the videos showing the model prediction vs the real environment in the case of a single model below. https://drive.google.com/open?id=1FzHQgosQNfbHsKXVewYrhuLrOq8jVEqH
> > >
> > > As commented in a different thread, we intend to open source our code, with the hope that other researcher can try our method on various other problem domains.

---

### Public Comment · (anonymous) · 2017-12-11
**questions about choice of horizon, and the changes to the original mujoco environment.**

Thanks for the interesting paper! However I have some questions about the choice of horizons. From your video, I notice for the hopper, it quickly falls over. I suspect it is because of your choice of short horizon (100 time step)? While I understand long time horizon would be difficult to optimize using BPTT, I assume it wouldn't be a problem for model free method like TRPO? And the changes to the original environment (using soft constraints instead of early termination) make it hard to compare against previous results (in terms of the video of performance, not pure score) like TRPO and PPO.

---

> ### Author Response · Authors · 2017-12-12
> **Re: questions about choice of horizon, and the changes to the original mujoco environment.**
>
> Thank you for your comment. We decided to work with horizon 100 mainly because of two reasons. First, we think that it is sufficient to demonstrate meaningful behaviors. Second, they are easier to compare the results in both BPTT and TRPO cases. Regarding the soft-constraint videos, we see similar behaviors in the videos of our methods and those of TRPO and PPO on the same setting (at the end it tries to dive to maximize the reward and curl up to make the center of mass even forward).

---

### Author Response · Authors · 2018-01-05
**Response**

We would like to thank all the reviewers for your comments. We really appreciate your feedback and we address your concerns below.

1. Our approach can be used with Bayesian neural networks or dropouts. In our experiments, we decided to use an ensemble of neural networks to model the dynamics because of its simplicity. We noticed that using different initializations and different sequences of training batches is enough to approximate the posterior distribution. This finding has also been shown in the work by Osband et. al., in which the best performance is attained by simply relying on different initializations for the ensemble of Q-functions, not requiring the sampling-with-replacement process that is prescribed by bootstrapping.

2. We provide the comparisons between BPTT and TRPO when a single model is employed. Thus, the only difference is the choice of policy optimizer. We made this point more explicit in the revised version.

3. We agree that the use of model ensembles can be further utilized for exploration. In this paper, our contribution makes the first step in this direction - showing that MB RL with model ensembles is competitive with model-free methods on state-of-the-art RL benchmarks, and can be used with expressive neural network models.

4. Even though it is true that, in simulation environments, the real-time complexity could be longer than model-free methods, the ultimate goal of model-based RL is to be able to use reinforcement learning in real-world robots, where the data collection is the bottleneck. Model ensembles can also be trained in a parallelizable way, which makes the training speed comparable to that of a single model.

5. To answer the question of further scaling, a more elaborate benchmark is required to fully understand the benefits and challenges of MB vs. MF, and this is something we are considering for future work.

6. Our method can be used with Q-learning for problems where a value function approach is desired. We have not tried this yet.

7. In the revised version, we have included longer training times for Ant and Humanoid, and fixed the reference to Depeweg et al. and Mishra et al. We also include the real-time complexity of our algorithms in the appendix.


References:
Osband, I., Blundell, C., Pritzel, A., and Van Roy, B., 2016. Deep exploration via bootstrapped DQN. In Advances in Neural Information Processing Systems (pp. 4026-4034).

Depeweg, S., Hernández-Lobato, J.M., Doshi-Velez, F. and Udluft, S., 2016. Learning and policy search in stochastic dynamical systems with Bayesian neural networks. arXiv preprint arXiv:1605.07127.

Mishra, N., Abbeel, P. and Mordatch, I., 2017. Prediction and Control with Temporal Segment Models. arXiv preprint arXiv:1703.04070.

---

### Decision · Program_Chairs · 2018-01-29
**ICLR 2018 Conference Acceptance Decision**

**Decision:**

Accept (Poster)

**Comment:**

The reviewers agree that the paper presents nice results on model based RL with an ensemble of models. The limited novelty of the methods is questioned by one reviewer and briefly by the others, but they all agree that this paper's results justify its acceptance.